# Determinants of Bone Health Status in a Multi-Ethnic Population in Klang Valley, Malaysia

**DOI:** 10.3390/ijerph17020384

**Published:** 2020-01-07

**Authors:** Chin Yi Chan, Shaanthana Subramaniam, Norazlina Mohamed, Soelaiman Ima-Nirwana, Norliza Muhammad, Ahmad Fairus, Pei Yuen Ng, Nor Aini Jamil, Noorazah Abd Aziz, Kok-Yong Chin

**Affiliations:** 1Department of Pharmacology, Universiti Kebangsaan Malaysia Medical Centre, Cheras 56000, Malaysia; chanchinyi94@gmail.com (C.Y.C.); shaanthana_bks@hotmail.com (S.S.); azlina@ppukm.ukm.edu.my (N.M.); imasoel@ppukm.ukm.edu.my (S.I.-N.); norliza_ssp@ppukm.ukm.edu.my (N.M.); 2Department of Anatomy, Universiti Kebangsaan Malaysia Medical Centre, Cheras 56000, Malaysia; apai.kie@gmail.com; 3Drug and Herbal Research Centre, Faculty of Pharmacy, Universiti Kebangsaan Malaysia Kuala Lumpur Campus, Jalan Raja Muda Abdul Aziz, Kuala Lumpur 50300, Malaysia; pyng@ukm.edu.my; 4Faculty of Health Science, Universiti Kebangsaan Malaysia Kuala Lumpur Campus, Jalan Raja Muda Abdul Aziz, Kuala Lumpur 50300, Malaysia; ainijamil@ukm.edu.my; 5Department of Family Medicine, Universiti Kebangsaan Malaysia Medical Centre, Cheras 56000, Malaysia; azah@ppukm.ukm.edu.my

**Keywords:** osteopenia, osteoporosis, bone mineral density, predictors, middle-aged, elderly

## Abstract

*Background and objectives:* Studies on osteoporosis risk factors are limited in Malaysia, so this study assesses the factors associated with bone health assessed using dual-energy X-ray absorptiometry (DXA) among Malaysians aged ≥40 years. *Subjects and Methods:* Data on demography, medical history, dietary and lifestyle practices of 786 Malaysians (51.4% women) aged ≥40 years recruited in Klang Valley were obtained. Their body composition and bone health were determined using DXA. The association between risk factors and bone health status was assessed using binary logistic regression. *Results:* The prevalence of suboptimal bone health and osteoporosis was higher in women (59.4% and 16.1%) than men (40.8% and 8.4%). Overall, the predictors of suboptimal bone health and osteoporosis among the subjects were increased age and higher fat mass. Lower monthly income was positively associated with osteoporosis. Being menopausal was a risk factor for both suboptimal bone health and osteoporosis in women. Women with no formal education were more likely to get osteoporosis. Being a smoker and Chinese were positively related to suboptimal bone health among men. Meanwhile, predictors of osteoporosis among men were regular alcohol and dairy product consumption, higher fat mass and having a tertiary education. *Conclusions:* This study calls for immediate and effective interventions for middle-aged and elderly populations with risk factors to halt the progression of bone loss.

## 1. Introduction

Osteoporosis is a progressive metabolic bone disease marked by loss of bone mass and micro-architectural deterioration of bone tissue which leaves the skeleton vulnerable to fracture [1,2]. Osteoporosis is asymptomatic until a fracture occurs, and the resulting complications pose significant burdens on the society and patients [3]. With the silver wave hitting developing countries like Malaysia, the prevalence and burden of osteoporosis will only grow larger [4]. However, data pertaining to osteoporosis in Malaysia are very limited. The most comprehensive study on hip fracture incidence among the elderly in Malaysia was carried out between 1996 and 1997 [5]. The study showed that the incidence of hip fracture was 88/100,000 in men and 218/100,000 in women during the study period. Additionally, the number of hip fracture cases in Malaysia was projected to increase by 3.55-fold from 2018 to 2050, which was the highest increase in the Asian region [6].

Studies have shown that risk factors associated with osteoporosis are dissimilar in different populations due to the inherent genetic and environmental variations. Identification of risk factors for osteoporosis may allow the relevant parties to strategize osteoporosis prevention efforts by targeting susceptible populations and modifying their lifestyles. This would help to minimize injury and disability, improve quality of life and reduce costs to society. In Malaysia, several studies have been conducted to determine the risk factors associated with osteoporosis in women [7,8]. A study involving 76 Malaysians postmenopausal women (45% Chinese, 42% Malay, 10% Indian and 3% other ethnic) indicated a positive association between hip bone mineral density (BMD) with parity, daily habitual tea consumption and body mass index (BMI); whereas duration of sleep was negatively associated with hip BMD [7]. Lim et al. (2005) found that increased age and years since menopause were positively associated with the occurrence of osteoporosis, while body weight, BMI and waist and hip circumference were negatively associated with osteoporosis among healthy Malaysian women (≥45 years) [8]. Another study found significant negative associations between bone ultrasound parameters and years since menopause, BMI and number of lifetime pregnancies in women ≥50 years [9]. Whereas in men, the studies in Malaysia are very limited. Several reports by Chin et al. established significant associations between endocrinological factors, body anthropometry and physical activity status with speed of sound, a bone quantitative sonometric index, in men ≥20 years [10,11,12,13,14]. From the pilot study previously conducted among Malaysian Chinese aged ≥40 years old, the prevalence of osteoporosis based on DXA was 15.3%. It was found that increased age and low monthly income elevated the risk of osteoporosis among them [15].

From the previous studies above, it is noted that studies on osteoporosis risk stratification based on DXA are limited in Malaysia. The available studies using DXA focuses on women so comparisons of risk factors between both sexes could not be conducted. Therefore, this study aimed to determine the predictors associated with bone health status assessed through DXA among Malaysians aged ≥40 years. This study involved both sexes and three main ethnic groups in Klang Valley, Malaysia, which is a highly urbanized area at the center of the Malaysian Peninsular. This region has a population demography of 45.9% Malay, 43.2% Chinese, 10.3% Indian and 0.6% other ethnic groups according to the latest census [16].

## 2. Materials and Methods

### 2.1. Study Design

The current study is a part of a larger population study using the same cohort of subjects, and the protocol of the study has been described in previous publications [15,17]. This cross-sectional study was conducted in Klang Valley, Malaysia from April 2018 to April 2019. Quota sampling with stratification based on sex (1:1) and ethnicity (45% Malay, 45% Chinese and 10% Indian and others) was used to recruit community-living Malaysians aged ≥40 years. The stratification resembled the population demography of Klang Valley, Malaysia [18]. The recruitment was advertised through mass media (local vernacular newspapers and radio broadcasts), as well as flyers and posters to community centers in Klang Valley, Malaysia. Subjects previously diagnosed with osteoporosis, metabolic bone diseases (Paget’s disease, osteogenesis imperfecta, osteomalacia and rickets), hypo/hyperparathyroidism, hypo/hypercalcemia, hypo/hyperthyroidism and/or who were receiving pharmacological treatment for osteoporosis (bisphosphonates, teriparatide, denosumab and strontium ranelate) or other treatments that could significantly impact bone metabolism (hormone-replacement therapy, sex hormone deprivation therapy, thiazide diuretics, anticonvulsants, antidepressants, glucocorticoids and thyroid supplements) were excluded. Those having mobility problems, needing a walking aid, having metal implants at the calcaneus, hip, spine, or femoral neck, suffered a lower limb fracture 2 years prior to the screening date, or a low impact fracture after the age of 50 years or those who could not complete the questionnaire or screening procedure were excluded as well. Potential participants were screened through a phone interview to determine their eligibility. Only subjects fulfilling the inclusion criteria were recruited and invited to the subsequent screening session, whereby they were briefed about the details of this project and provided written informed consent to participate in this study. The study protocol had been reviewed and approved by the Ethics Committee of Universiti Kebangsaan Malaysia Medical Centre (approval code: UKM PPI/111/8/JEP 2017-721).

### 2.2. Subjects

During the screening session, the subjects answered a questionnaire regarding demographic details, medical history, diet and lifestyle practices during the interview session. The age of the subjects was derived from information on their identification cards. Ethnicity, occupation, sex, menstrual status, age of menarche, age of menopause, parity and presence of pre-existing medical conditions and medical treatments were self-declared. Subjects were grouped based on their age, wherein those aged 40–59 years were referred to as “middle-aged” [19,20] while those aged ≥60 years were referred to as “elderly” [21]. The occupation of the subjects was classified into manual or sedentary based on the amount of time they spent on walking, carrying heavy objects or sitting at the workplace or in a vehicle. The sedentary group included retirees, housewives and subjects in job transition. Based on household income, the subjects were categorized into the bottom 40% (B40, with household income < Ringgit Malaysia (RM) 7640), the middle 40% (M40, with household income RM 7640–15,159) and the top 20% (T20, with household income >RM 15,160) as per the Malaysian census [22].

### 2.3. Dietary and Lifestyle Practices

The subjects disclosed their smoking behavior, intake of dairy products (milk, yogurt and cheese), beverages (coffee, tea and alcohol—beer, wine or spirits) and calcium supplement intake [9]. The determination of regular consumers of the products above had been detailed in the prior publication [15].

### 2.4. Physical Activity Assessment

The International Physical Activity Questionnaire (IPAQ-short form) was used to assess the physical activity level of the subjects [23]. It requires the subjects to recall the average amount of time spent in high-intensity activity, moderate-intensity activity, walking and sitting/lying down (except sleeping) in a week. The physical activity level of the subjects was classified as inactive, minimally active or HEPA-active (health-enhancing physical activity) based on the total metabolic equivalent of task (MET) score (converted from the time spent in each activity) or other additional criteria [24]. The validity and reliability of IPAQ have been tested in the Malaysian population [25].

### 2.5. Body Anthropometry Measurements

A stadiometer (Seca, Hamburg, Germany) was used to measure the standing height of the subjects (to the nearest 1 cm) without shoes. A weighing scale (Tanita, Tokyo, Japan) was used to measure the bodyweight of the subjects (to the nearest 0.1 kg) with light clothing. The BMI was calculated as the ratio of weight in kg to the square of height in meter. Generally, for subjects below 65 years old, BMI < 18.5 kg/m^2^ were underweight, 18.5–24.9 kg/m^2^ were normal, 25.0–29.9 kg/m^2^ were overweight and >30.0 kg/m^2^ were obese [26]. For subjects above 65 years old, 22–27 kg/m^2^ were normal, >27 kg/m^2^ were overweight and <22 kg/m^2^ were underweight [27]. A soft measuring tape was used to measure the waist circumference between the lowest rib margin and the iliac crest of the subjects (to the nearest 0.1 cm).

### 2.6. Bone Mineral Density Assessment

Bone mineral density (BMD) of the lumbar spine (L1–L4) and the hip (femoral neck and total hip) of the non-dominant leg was measured with the Hologic Discovery QDR Wi densitometer, DXA (Hologic, MA, USA). Daily calibration of the device was performed using the phantom supplied by the manufacturer. The short term in-vivo coefficient of variation for the DXA machine was 1.8% and 1.2% for the lumbar spine and total hip, respectively. Body composition analysis was also performed using the same DXA device, which generates body fat percentage and lean body mass. The T-score was also computed by the DXA software by comparing the BMD values of the subjects against the reference values of the Singaporean population. The diagnosis of osteoporosis/osteopenia was done based on the T-scores values; a T-score of ≤−2.5 indicates osteoporosis, between −2.5 and −1 indicates osteopenia, and >−1 indicates normal bone health [28].

### 2.7. Statistical Analysis

Statistical analyses were performed using the Statistical Package for Social Science version 22.0 (IBM, Armonk, NY, USA). Kolmogorov–Smirnov test was used to assess the normality of the data. The relationship between the bone health status of the subjects and predictors of interest was determined using multivariate logistic regression analysis. All continuous variables and dummy-coded categorical predictors (ethnicity, education level, physical activity status, parity and menstrual status) were force-entered in the logistic regression. In the model for predictors of suboptimal bone health, subjects with normal bone health status were coded as “0”, while subjects with osteoporosis and osteopenia were coded as “1”. In the model for predictors of osteoporosis, subjects with normal bone health or osteopenia were coded as “0”, while subjects with osteoporosis were coded as “1”. The dichotomized bone health status was entered as the dependent variable in the logistic regression. The effect size of each predictor was expressed as odds ratio (OR) and 95% confidence interval (CI). A *p*-value < 0.05 was considered to be statistically significant.

## 3. Results

Overall, 910 subjects participated the study, but 124 were not eligible because they were thiazide diuretic users (n = 20), glucocorticoid users (n = 32), receiving cancer treatment (n = 4), having mobility problems (n = 12), hysterectomy before menopause (n = 5), hormone medication users (n = 30) (13 for hormone replacement therapy, 2 for sex hormone deprivation therapy and 15 for thyroid supplements) and did not complete the study procedures (n = 21). Finally, data from the remaining 786 subjects, consisting of 382 men and 404 women, were included in the analysis. The mean age of the subjects was 57.16 (SD = 9.12) years. Of the female subjects, 265 were menopausal (average years since menopause = 9.68 (SD = 6.68)). The ethnic composition of the subjects was 46.2% Chinese, 43.5% Malays and 10.3% Indians or other ethnic groups. The subjects predominantly resided in Hulu Langat district in Klang Valley (79.5%), were married (93.4%), had sedentary jobs (94.1%) and belonged to the B40 group (93.1%). Most of the subjects were secondary school graduates (41.9%), not regular consumers of dairy products (62.6%) and calcium supplements (85.0%), non-smokers (77.7%) and non-alcohol drinkers (87.8%). They consumed coffee or tea regularly (81.4%). Only 12% of the subjects were HEPA-active. Regarding the history of previous fractures, 11 subjects indicated previous fractures due to motor vehicle accidents and 8 subjects due to falls. Overall, the prevalence of osteopenia and osteoporosis among the subjects was 38.0% and 12.3%, respectively. Based on sex, the prevalence of osteoporosis was 8.4% for men and 16.1% for women. Since the characteristics of the subjects have been described in previous reports [17], they are not tabulated in this paper.

The current study revealed that increased age (OR: 1.044, 95% CI: 1.022–1.066, *p* ≤ 0.001) and fat mass (OR: 1.000228, 95% CI: 1.000090–1.000366, *p* = 0.001) positively predicted suboptimal bone health among the subjects. Meanwhile, Indian or other ethnicities (vs. Chinese, OR = 0.433, 95% CI: 0.236–0.793, *p* = 0.007) and higher body weight (OR: 0.792, 95% CI: 0.700–0.896, *p* ≤ 0.001) were negatively associated with suboptimal bone health. Sub-analysis based on sex showed that Malay (vs. Chinese, OR: 0.454, 95% CI: 0.245–0.841, *p* = 0.012) or Indian and others (vs. Chinese, OR: 0.361, 95% CI: 0.150–0.871, *p* = 0.023) were negatively associated with suboptimal bone health in men. On the other hand, being a smoker (OR: 1.741, 95% CI: 1.003–3.022, *p* = 0.049) was positively related with suboptimal bone health among men. For women, being menopausal (vs. pre-menopausal, OR: 3.433, 95% CI: 1.412–8.347, *p* = 0.006) was associated positively with suboptimal bone health. A higher body weight (OR = 0.832, 95% CI = 0.722–0.958, *p* = 0.010) was associated negatively with suboptimal bone health in women (Table 1).

The logistic regression model also showed that increased age (OR: 1.096, 95% CI: 1.059–1.135, *p* ≤ 0.001), low income status (vs. high income status, OR: 4.031, 95% CI: 1.422–11.430, *p* = 0.004) and higher fat mass (OR: 1.000324, 95% CI: 1.000107–1.000542, *p* ≤ 0.001) were associated positively with osteoporosis in the overall subjects. A higher body weight (OR: 0.729, 95% CI: 0.606–0.878, *p* = 0.001), being men (OR: 0.189, 95% CI: 0.057–0.623, *p* = 0.006) and minimally-active (vs. inactive, OR: 0.494, 95% CI: 0.281–0.866, *p* = 0.014) were negatively associated with osteoporosis among the subjects. Sub-analysis based on sex revealed that older age positively predicted osteoporosis among men (OR: 1.123, 95% CI: 1.045–1.207, *p* = 0.002). In addition, regular alcohol consumption (vs. non-drinker, OR: 4.146, 95% CI: 1.432–13.616, *p* = 0.010), regular dairy product consumption (vs. non-drinker, OR: 3.167, 95% CI: 1.003–10.001, *p* = 0.049) and having at least an undergraduate degree (vs. no formal/primary education, OR: 8.272, 95% CI: 1.315–52.027, *p* = 0.024) and higher fat mass (OR: 1.000641, 95% CI: 1.000149–1.001133, *p* = 0.011) were positively associated with osteoporosis in men. In contrast, being minimally-active (vs. inactive, OR: 0.381, 95% CI: 0.148–0.983, *p* = 0.046) was associated negatively with osteoporosis in men. Whereas in women, menopause (vs. pre-menopause, OR: 10.795, 95% CI: 1210–96.282, *p* = 0.033) was associated positively with osteoporosis. An education level of ≥secondary school (vs. no formal education/primary; secondary OR: 0.384, 95% CI: 0.135–1.095, *p* = 0.073; diploma OR: 0.217, 95% CI: 0.057–0.825, *p* = 0.025) was associated negatively with osteoporosis among women (Table 2).

## 4. Discussion

The current study showed that osteoporosis can occur in populations without major secondary risk factors of bone loss. Overall, 8.4% of the subjects recruited suffered from osteoporosis and 32.5% had osteopenia. The prevalence values in this study are similar to the prevalence of osteopenia (men: 31.5%, women: 41.9%) and osteoporosis (men: 9.7%, women: 15.4%) in Korean population ≥40 years [29]. In an older Korean cohort (aged >50 years), a higher prevalence of osteoporosis among women (38.0%) compared to men (7.3%) was also reported [30]. Being men was also negatively associated with osteoporosis in this study. The greater prevalence of osteoporosis among women could be attributed to accelerated bone loss due to estrogen deficiency after menopause [31] and a lower peak bone mass compared to men [32]. An earlier study found a higher prevalence of osteoporosis among Malaysian men (10.6%) than women (8.0%), but the subjects were sampled at a university hospital and older (aged ≥50 years). Besides, they used a quantitative bone sonometer to estimate the bone health status of the subjects instead of DXA [33].

Increased age is the most important risk factor of bone loss for both men and women. Women experience accelerated bone loss due to menopause. The resultant estrogen deficiency leads to increased bone resorption and decreased bone formation [34]. This also explains the menopausal state as the major determinant of bone health in women in this study. Some reports postulated that the elevated follicle-stimulating hormone (FSH) level during the menopausal transition could be harmful to the skeleton but the effects pale compared to the estrogen deficiency [35]. After menopause, gradual bone loss continues to occur in women due to aging [36]. The gradual decline of bone health in men is contributed by testosterone deficiency syndrome and aging [37,38,39]. Aging also causes a disturbance of other hormones, including thyroid hormones [35], insulin-like growth factor-1 [11] and parathyroid hormones [40], as well as transient low-grade inflammation [41] which predispose an individual to osteoporosis. However, the endocrinological factors associated with osteoporosis were not examined in the current studies.

Chinese ethnicity was associated positively with suboptimal bone health among men in this study. Distinct ethnic differences in hip fracture incidence among Malaysians (>50 years) had been reported previously, whereby it was the highest among Chinese (27.4%), followed by 18.2% Indians and 16.9% Malays [5]. A study among postmenopausal women (>50 years) in a Malaysian hospital also indicated that the prevalence of low hip BMD (osteoporosis or suboptimal bone health) was the highest among Chinese (62%), followed by Malay (26%), Indians (10%) and other ethnic groups (2%) [7]. The reason for this observation could be multifactorial, i.e., the lack of calcium in the Chinese diet [42] and other previously unknown genetic variations.

Lower education backgrounds were positive predictors of osteoporosis in women of this study. This is similar to a study among postmenopausal Chinese women, which found that higher education levels were independently associated with a lower prevalence of osteoporosis [43]. In contrast, men with higher education were associated with a higher risk of osteoporosis in this study. They might be sedentary office workers who spent less time in physical activities. This observation was in contrast with other studies that reported that lower education levels were a positive predictor of osteoporosis in men [44,45].

Lower monthly income was positively associated with osteoporosis in this study. In the current study, 93.1% of subjects in this study belong to the low socioeconomic group (B40) according to the Malaysian census, who might face limited access to healthcare knowledge, professionals and facilities. They might not able to afford healthy food, supplements, the luxury of time for exercise and medical consultation [46].

Being moderately active was negatively associated with osteoporosis in this study. Many studies have confirmed the role of physical activity on bone health [47,48,49,50]. Moderate-intensity aerobic training exerts significant positive effects on bone formation and bone density while decreasing bone resorption, subsequently delaying the progression of bone loss [51]. Aerobic exercises are particularly effective in stimulating the activity of osteoblasts [52]. Exercise increases the thickness and resistance of cortical bone at loaded skeletal sites among the elderly [53]. Preservation of bone strength among the exercising elderly is attributed to a slower loss of endocortical bone and an increase in tissue density, independent of bone size.

Alcohol consumption was a strong positive predictor of osteoporosis among men in this study. This study agreed with some findings that alcohol consumption was negatively associated with BMD and positively with fractures [54,55]. Experimental studies prevailingly demonstrated the negative effect of alcohol on bone cells and animals [56]. Alcohol consumption depletes calcium reserves, damages the pancreas, leading to low vitamin D synthesis and poor calcium absorption. Chronic alcohol intake destroys bone mass and reduces bone development, causing bone to be prone to fissure formation in humans [57]. However, several studies showed that moderate alcohol consumption is not harmful, or even beneficial for bone [58,59]. Moderate alcohol use was associated with increased BMD in some studies, which might be attributable to higher endogenous estrogen levels among moderate drinkers [60,61,62]. The effects of alcohol on bone might be dose-dependent, but the current study did not consider this aspect during the analysis.

Adequate dietary calcium intake through dairy sources is a well-recognized osteoprotective behavior [63]. From our previous study, most subjects were aware of the importance of sufficient calcium intake in maintaining bone health [64]. However, a positive association between dairy product intake and osteoporosis was found among men in this study. This might reflect that men with prior knowledge of their bone health have begun to increase the intake of dairy products. Hence, this observation may be casual rather than causal. Sufficient intake of calcium (1000–1200 mg/day) through diet or supplements has been recommended for older individuals to prevent osteoporosis [65]. However, the intake of calcium supplements was not significantly associated with bone health in this study. Since most subjects were not regular calcium supplements users, this would attenuate the relationship.

Smoking was found to be strongly associated with suboptimal bone health among men in this study. Nicotine in cigarettes is harmful to the bone [63] and cigarette smoking was associated with low BMD in several epidemiological studies [66,67,68]. It has been predicted that tobacco smoke influences bone mass indirectly through alteration of body weight, parathyroid hormone-vitamin D axis, adrenal hormones, sex hormones and increased oxidative stress on bone tissues [68].

The relationship between parity and bone health remains unclear because positive, negative or nil associations have been reported [69,70,71]. Several studies found that parity protected bone health in women, whereby bone loss was slower in multiparous women compared to nulliparous women [72,73]. However, bone health was not significantly associated with parity among women in this study. Another positive predictor of suboptimal bone health and osteoporosis in this study was increased fat mass. This finding is consistent with previous studies, which suggested that excessive fat mass was associated with decreased bone mass [74,75,76,77]. This observation may be linked to a chronic inflammatory response and abnormal cytokine production induced by adiposity. Proinflammatory cytokines, such as interleukin-1, interleukin-6 and tumor necrosis factor-alpha, stimulate the differentiation of osteoclasts that govern bone resorption, thus contributing to bone loss [78]. Furthermore, adipose tissue could sequester vitamin D and other lipophilic hormones important to bone health, thus decreasing their bioavailability and preventing them from reaching the skeletal tissues [79].

Increased body weight was negatively associated with suboptimal bone health and osteoporosis in this study. They increase mechanical loading on the bone, thus encouraging it to undergo adaptive changes to support the increased load [80]. Extensive data have shown that body weight and lean mass are important determinants of BMD [81,82,83,84]. Osteocytes have been suggested to act as the mechanosensor of the skeletal system [85]. They respond to increased mechanical loading by sending signals to other bone cells to reduce osteoclastic bone resorption or increase osteoblastic bone formation, thereby increasing bone mass and strength [86]. Other researchers also indicated that increased body weight may trigger the increased secretion of pancreatic hormones that promote bone homeostasis and formation [87].

This study has several limitations and the findings need to be interpreted with caution. Causality between osteoporosis and its associated risk factors cannot be inferred in this study due to the nature of its cross-sectional design. A prospective study is needed to confirm the relationship between the risk factors of interest and bone health. Furthermore, the intake of calcium-rich foods, dairy products, beverages and calcium supplements, as well as smoking were not studied in depth using specialized questionnaires. So, a dose–response relationship between the consumption of these products and bone health could not be derived. Moreover, the subjects recruited were healthier than the general population because those with major risk factors of osteoporosis were excluded. The vitamin D level and sun exposure of the subjects were not determined in this study, thus their association with bone health cannot be inferred. Despite these limitations, this study provides important information for policymakers in planning strategies to prevent osteoporosis and its associated problems among the middle-aged and elderly population in Malaysia.

## 5. Conclusions

In conclusion, the prevalence of suboptimal bone health and osteoporosis among Malaysians aged ≥40 years is substantial. The Chinese ethnic group and women are more susceptible to suboptimal bone health and osteoporosis. The positive predictors of suboptimal bone health and osteoporosis among the subjects are increased age and higher fat mass. Lower monthly income is positively associated with osteoporosis. Among women, being menopausal is a risk factor for both suboptimal bone health and osteoporosis. Women with no formal education more likely to get osteoporosis. Among men, being a smoker and Chinese were positively related to suboptimal bone health. On the other hand, regular alcohol and dairy consumption, being a smoker and having a higher education level (at least a degree and above) were positively associated with osteoporosis. An osteoporosis prevention program focusing on the modifiable risk factors may help to decrease the burden of suboptimal bone health and osteoporosis among Malaysian middle-aged and elderly populations.

## Figures and Tables

**Table 1 ijerph-17-00384-t001:** Predictors of suboptimal bone health of the study population.

Variables	Odds Ratio (OR)	95% CI for OR		*p*-Value
		Lower	Upper	
**Overall (among both sexes)**				
Age (Years)	1.044	1.022	1.066	**≤0.001**
Sex *(Men vs. Women (ref.))*	0.485	0.227	1.033	0.061
Ethnicity *Malays vs. Chinese (ref.)*	0.687	0.449	1.051	0.083
*Indian and others vs. Chinese (ref.)*	0.433	0.236	0.793	**0.007**
Monthly income *(B40 vs. M40 + T20 (ref.))*	1.153	0.577	2.302	0.688
Nature of jobs *(Sedentary vs. Manual (ref.))*	0.797	0.389	1.632	0.534
Education level *Secondary vs. No formal education/Primary (ref.)*	0.916	0.485	1.730	0.788
*Diploma vs.* *No formal education/Primary (ref.)*	0.906	0.450	1.824	0.782
*Degree and above vs. No formal education/Primary (ref.)*	0.794	0.390	1.617	0.526
Height (cm)	1.025	0.991	1.060	0.148
Weight (kg)	0.792	0.700	0.896	**≤0.001**
Fat Mass (kg)	1.000228	1.000090	1.000366	**0.001**
Lean Mass (kg)	1.000065	0.999941	1.000189	0.308
Waist Circumference (cm)	1.019	0.995	1.044	0.123
Smoking habits *(Smokers vs. Non-smokers (ref.))*	1.490	0.929	2.388	0.98
Alcohol drinking *(Drinker vs. Non-drinker (ref.))*	1.164	0.683	1.986	0.576
Dairy product *(Drinker vs. Non-consumer (ref.))*	1.038	0.733	1.471	0.833
Coffee or tea intake *(Drinker vs. Non-drinker (ref.))*	1.000	0.652	1.535	0.999
Calcium supplement intake *(User vs. Non-user (ref.))*	1.215	0.768	1.920	0.405
Physical activity *Minimally-active vs. Inactive (ref.)*	0.785	0.544	1.003	0.195
*HEPA-active vs. Inactive (ref.)*	0.580	0.335	1.066	0.051
**Sub-analysis among men**				
Age (Years)	0.995	0.967	1.025	0.747
Ethnicity *Malays vs. Chinese (ref.)*	0.454	0.245	0.841	**0.012**
*Indian and others vs. Chinese (ref.)*	0.361	0.150	0.871	**0.023**
Monthly income *(B40 vs. M40 + T20 (ref.))*	0.504	0.194	1.314	0.161
Nature of jobs *(Sedentary vs. Manual (ref.))*	0.421	0.152	1.162	0.095
Education level *Secondary vs. No formal education/Primary (ref.)*	1.054	0.425	2.618	0.909
*Diploma vs. No formal education/Primary (ref.)*	1.148	0.434	3.037	0.781
*Degree and above vs. No formal education/Primary (ref.)*	1.109	0.414	2.968	0.837
Height (cm)	1.046	0.996	1.099	0.074
Weight (kg)	0.799	0.622	1.026	0.079
Fat Mass (kg)	1.000198	0.999930	1.000465	0.148
Lean Mass (kg)	1.000044	0.999785	1.000303	0.739
Waist Circumference (cm)	1.023	0.985	1.061	0.238
Smoking habits *(Smokers vs. Non-smokers (ref.))*	1.741	1.003	3.022	**0.049**
Alcohol drinking *(Drinker vs. Non-drinker (ref.))*	1.161	0.625	2.158	0.637
Dairy product *(Drinker vs. Non-consumer (ref.))*	0.830	0.483	1.427	0.501
Coffee or tea intake *(Drinker vs. Non-drinker (ref.))*	0.929	0.449	1.920	0.841
Calcium supplement intake *(User vs. Non-user (ref.))*	1.512	0.699	3.267	0.293
Physical activity *Minimally-active vs. Inactive (ref.)*	0.770	0.457	1.297	0.326
*HEPA-active vs. Inactive (ref.)*	0.686	0.328	1.435	0.317
**Sub-analysis among women**				
Age (Years)	1.044	0.993	1.097	0.092
Age of menarche (Years)	0.928	0.808	1.067	0.293
Parity *1–3 pregnancies vs. zero pregnancy (ref.)*	0.612	0.293	1.278	0.191
*>3 pregnancies vs. zero pregnancy (ref.)*	0.550	0.261	1.158	0.115
Current menstrual status *Peri-menopause vs. Pre-menopause (ref.)*	1.381	0.563	3.389	0.481
*Post-menopause vs. Pre-menopause (ref.)*	3.433	1.412	8.347	**0.006**
Ethnicity *Malays vs. Chinese (ref.)*	1.045	0.519	2.103	0.902
*Indian and others vs. Chinese (ref.)*	0.571	0.216	1.515	0.260
Monthly income *(B40 vs. M40 + T20 (ref.))*	3.318	0.824	13.365	0.092
Nature of jobs *(Sedentary vs. Manual (ref.))*	1.310	0.393	4.362	0.660
Education level *Secondary vs. No formal education/Primary (ref.)*	0.649	0.242	1.742	0.391
*Diploma vs. No formal education/Primary (ref.)*	0.475	0.155	1.450	0.191
*Degree and above vs. No formal education/Primary (ref.)*	0.443	0.143	1.370	0.158
Height (cm)	1.003	0.949	1.059	0.919
Weight (kg)	0.832	0.722	0.958	**0.010**
Fat Mass (kg)	1.000028	0.999897	1.000160	0.672
Lean Mass (kg)	1.000161	0.999992	1.000331	0.062
Waist Circumference (cm)	1.016	0.981	1.052	0.380
Smoking habits *(Smokers vs. Non-smokers (ref.))*	0.995	0.245	4.046	0.995
Alcohol drinking *(Drinker vs. Non-drinker (ref.))*	1.188	0.352	4.006	0.781
Dairy product *(Drinker vs. Non-consumer (ref.))*	1.261	0.759	2.095	0.371
Coffee or tea intake *(Drinker vs. Non-drinker (ref.))*	0.963	0.540	1.719	0.899
Calcium supplement intake *(User vs. Non-user (ref.))*	1.187	0.637	2.211	0.589
Physical activity *Minimally-active vs. Inactive (ref.)*	0.950	0.537	1.679	0.860
*HEPA-active vs. Inactive (ref.)*	0.426	0.173	1.045	0.062

The bolded *p*-values are statistically significant. The odds ratio was obtained through a multivariate logistic regression model. The predictors are adjusted to each other; ref.: reference group.

**Table 2 ijerph-17-00384-t002:** Predictors of osteoporosis of the study population.

Variables	Odds Ratio (OR)	95% CI for OR		*p*-Value
		Lower	Upper	
**Overall (among both sexes)**				
Age (Years)	1.096	1.059	1.135	**≤0.001**
Sex *(Men vs. Women (ref.))*	0.189	0.057	0.623	**0.006**
Ethnicity *Malays vs. Chinese (ref.)*	0.853	0.449	1.621	0.627
*Indian and others vs. Chinese (ref.)*	0.683	0.253	1.849	0.178
Monthly income *(B40 vs. M40 + T20 (ref.))*	4.031	1.422	11.430	**0.009**
Nature of jobs *(Sedentary vs. Manual (ref.))*	0.235	0.029	1.963	0.941
Education level *Secondary vs. No formal education/Primary (ref.)*	0.655	0.303	1.414	0.281
*Diploma vs. No formal education/Primary (ref.)*	0.596	0.234	1.517	0.278
*Degree and above vs. No formal education/Primary (ref.)*	1.004	0.388	2.597	0.994
Height (cm)	1.014	0.963	1.068	0.860
Weight (kg)	0.729	0.606	0.878	**0.001**
Fat Mass (kg)	1.000324	1.000107	1.000542	**0.004**
Lean Mass (kg)	1.000077	0.999904	1.000251	0.383
Waist Circumference (cm)	1.003	0.969	1.038	0.382
Smoking habits *(Smokers vs. Non-smokers (ref.))*	1.032	0.445	2.393	0.113
Alcohol drinking *(Drinker vs. Non-drinker (ref.))*	1.963	0.852	4.522	0.620
Dairy product *(Drinker vs. Non-consumer (ref.))*	1.144	0.671	1.951	0.618
Coffee or tea intake *(Drinker vs. Non-drinker (ref.))*	0.856	0.464	1.579	0.761
Calcium supplement intake *(User vs. Non-user (ref.))*	1.103	0.587	2.070	0.588
Physical activity *Minimally-active vs. Inactive (ref.)*	0.494	0.281	0.866	**0.014**
*HEPA-active vs. Inactive (ref.)*	0.846	0.371	1.928	0.691
**Sub-analysis among men**				
Age (Years)	1.123	1.045	1.207	**0.002**
Ethnicity *Malays vs. Chinese (ref.)*	0.617	0.174	2.191	0.456
*Indian and others vs. Chinese (ref.)*	0.459	0.068	3.074	0.422
Monthly income *(B40 vs. M40 + T20 (ref.))*	3.780	0.599	23.868	0.157
Nature of jobs *(Sedentary vs. Manual (ref.))*	0.400	0.153	1.139	0.095
Education level *Secondary vs. No formal education/Primary (ref.)*	3.194	0.605	16.874	0.171
*Diploma vs. No formal education/Primary (ref.)*	5.986	0.954	37.542	0.056
*Degree and above vs. No formal education/Primary (ref.)*	8.272	1.315	52.027	**0.024**
Height (cm)	1.097	0.995	1.210	0.062
Weight (kg)	0.528	0.338	0.825	**0.005**
Fat Mass (kg)	1.000641	1.000149	1.001133	**0.011**
Lean Mass (kg)	1.000255	0.999831	1.000679	0.239
Waist Circumference (cm)	1.006	0.943	1.074	0.847
Smoking habits *(Smokers vs. Non-smokers (ref.))*	2.441	0.789	7.552	0.122
Alcohol drinking *(Drinker vs. Non-drinker (ref.))*	4.416	1.432	13.616	**0.010**
Dairy product *(Drinker vs. Non-consumer (ref.))*	3.167	1.003	10.001	**0.049**
Coffee or tea intake *(Drinker vs. Non-drinker (ref.))*	0.403	0.100	1.613	0.199
Calcium supplement intake *(User vs. Non-user (ref.))*	0.587	0.145	2.372	0.455
Physical activity *Minimally-active vs. Inactive (ref.)*	0.319	0.113	0.898	0.031
*HEPA-active vs. Inactive (ref.)*	0.406	0.074	2.225	0.299
**Sub-analysis among women**				
Age (Years)	1.055	0.999	1.114	0.054
Age of menarche (Years)	0.919	0.747	1.130	0.423
Parity *1–3 pregnancies vs. zero pregnancy (ref.)*	1.169	0.436	3.135	0.757
*>3 pregnancies vs. zero pregnancy (ref.)*	1.584	0.575	4.367	0.374
Current menstrual status *Peri-menopause vs. Pre-menopause (ref.)*	3.972	0.311	50.682	0.288
*Post-menopause vs. Pre-menopause (ref.)*	10.795	1.210	96.282	**0.033**
Ethnicity *Malays vs. Chinese (ref.)*	1.063	0.434	2.604	0.893
*Indian and others* *vs.* *Chinese (ref.)*	0.511	0.139	1.876	0.312
Monthly income *(B40 vs. M40 + T20 (ref.))*	3.893	0.901	16.821	0.069
Nature of jobs *(Sedentary vs. Manual (ref.))*	0.657	0.069	6.278	0.715
Education level *Secondary vs. No formal education**/Primary (ref.)*	0.384	0.135	1.095	**0.073**
*Diploma vs. No formal education/Primary (ref.)*	0.217	0.057	0.825	**0.025**
*Degree and above vs. No formal education/Primary (ref.)*	0.393	0.097	1.589	0.190
Height (cm)	0.971	0.904	1.043	0.426
Weight (kg)	0.810	0.656	1.001	0.051
Fat Mass (kg)	1.000202	0.999938	1.000467	0.133
Lean Mass (kg)	1.000015	0.999833	1.000196	0.873
Waist Circumference (cm)	0.991	0.950	1.033	0.664
Smoking habits *(Smokers vs. Non-smokers (ref.))*	0.995	0.245	4.046	0.995
Alcohol drinking *(Drinker vs. Non-drinker (ref.))*	0.318	0.030	3.415	0.344
Dairy product *(Drinker vs. Non-consumer (ref.))*	1.021	0.535	1.951	0.949
Coffee or tea intake *(Drinker vs. Non-drinker (ref.))*	0.933	0.454	1.915	0.850
Calcium supplement intake *(User vs. Non-user (ref.))*	1.038	0.476	2.262	0.926
Physical activity *Minimally-active vs. Inactive (ref.)*	0.644	0.301	1.376	0.256
*HEPA-active vs. Inactive (ref.)*	0.993	0.335	2.940	0.989

The bolded *p*-values are statistically significant. The odds ratio was obtained through a multivariate logistic regression model. The predictors are adjusted to each other; ref: reference group.

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
