# Peer review of "Determinants of Bone Health Status in a Multi-Ethnic Population in Klang Valley, Malaysia"

_ijerph, 2020, doi:10.3390/ijerph17020384_

Round 1

Reviewer 1 Report

There is little existing data on rates of osteoporosis and osteopenia in the Malaysian population, especially for men. This subset of a larger cross sectional study deliberately designed to reflect the Malaysian demographics in adults over 40 years demonstrates the expected higher rate of osteopenia and osteoporosis in women, and explores the relationship between bone density and previously described risk factors by multivariate analysis. A few additional questions follow. Subjects: 1. I note from reference 17 that there were specific exclusions based on oestrogen therapy and bisphosphonate therapy as well as those mentioned in the text. Because this is so relevant to the BMD data this should be more explicit in this paper. 2. Was previous fracture history captured in the pre-existing medical conditions? If so was there enough data for this to be a variable? 3. Why are there different classifications of normal weight and overweight in the younger and older age groups? 4. What was the reference population used to determine the T-scores which were later used to classify subjects as osteopenic or osteoporotic? Important as this population's mean height from reference 17 is significantly lower than Caucasian reference populations. Data analysis 5. Distribution of subjects: Mean and SD is quoted universally. Was the data normally distributed? 6. Was there any relationship between BMD and prior fracture history? 7. As the Chinese ethnic group had lower BMD as a group, was this dependent on any other covariates eg age or height? General comments Several of the references are incomplete (eg ref 17) in giving only volume of journal but not page number.

Author Response

Dear reviewer, 

Thank you for reviewing our manuscript. We appreciated your constructive comments and are responded in the file attached.

Thank you 

Reviewer 2 Report

This study aimed to assess the factors associated with bone health status assessed using dual-energy X-ray absorptiometry (DXA) among Malaysians aged ≥ 40 years. A total of 786 Malaysians (382 men, 404 women) aged ≥ 40 years were recruited using quota sampling technique. Information on subjects’ demography, medical history, physical activity status, dietary and lifestyle practices was obtained. The authors found that the prevalence of suboptimal bone health and osteoporosis were higher in women than men. Overall, the predictors of suboptimal bone health and osteoporosis were increased age and higher fat mass. Subgroup analysis based on sex showed that being menopausal and having higher fat mass were risk factors for both suboptimal bone health and osteoporosis in women. Regular alcohol consumption and tertiary education (vs no formal/primary education) in men, and low income status (vs high income status) in women were also positively associated with osteoporosis. Therefore, they concluded that immediate and effective intervention approaches were needed for middle-aged and elderly populations with risk factors to halt the progression of bone loss. However, this study is merely a survey of risk factors for osteoporosis in a region without contrast comparisons or detailed discussion for the possible underlying mechanisms. The findings were common senses in the field of osteoporosis. Moreover, there is a bottom of articles describing such risk factors. Therefore the manuscript is less creative and may add not much to current knowledge.

My comments are as follows:

In the section of Method, there is a lack of a survey regarding important confounders (such as the use of hormone replacement therapy, sun exposure on the skin for vitamin D absorption) for osteoporosis. Lack of considering these factors will largely confound this result of the study. For every participant, the authors should consider to use a special questionnaire (there are many nutrition survey questionnaire available) for surveying the nutritional status, which is more relative to the calcium status and therefore the occurrence of osteopenia and osteoporosis. Does the study exclude participants with pre-existing bone pathology? The style of references is inconsistent. Please make changes for your references to conform to the journal’s style.

Author Response

Dear reviewer, 

Thank you for reviewing our manuscript. We appreciated your constructive comments and are responded in the file attached. 

Thank you. 

Reviewer 3 Report

The manuscript investigates the bone health of a multi-ethnic population in Klang Valley, Malaysia by answering questionnaires and measuring bone mineral density, which involved men and women and the three main races lived in that region. The authors found that increased age and higher fat mass were the most important risk factors for bone loss in men and women. In addition, menopause is also an important risk factor for bone loss in women. Among races, the authors also point out that the Chinese race is positively correlated with poor bone health in men. The manuscript is easy to follow and the conclusions are convincing and interesting. This study is beneficial to effective interventions to prevent bone loss in the elderly population.

Comment:

Without seeing the raw data, the authors can provide the raw data in a supplementary way.

Mini comment:

Line 173: need space between “the” and “B40”

Line 196: higher fat mass “CI: 1.000131-1.000378, p=0.005” p value is not match table “”p ≤ 0.001”

Line 210: 0.122? or 0.337? It is different between line 210 and table 2.

Line 464: repeated the article no. e0205045.

The last page number of the referenced article should be marked uniformly, xxx-xxx or xxx-x.

Author Response

(The authors gave the same response as above.)
